# Just add sugar for carbohydrate induced self-assembly of curcumin

Sandy Wong [1], Jiacheng Zhao[1], Cheng Cao[1], Chin Ken Wong [1], Rhiannon P. Kuchel[2], Sergio De Luca[3], James M. Hook[1,2], Christopher J. Garvey[4], Sean Smith[3,5], Junming Ho [1] & Martina H. Stenzel [1]

In nature, self-assembly processes based on amphiphilic molecules play an integral part in the design of structures of higher order such as cells. Among them, amphiphilic glycoproteins or glycolipids take on a pivotal role due to their bioactivity. Here we show that sugars, in particular, fructose, are capable of directing the self-assembly of highly insoluble curcumin resulting in the formation of well-defined capsules based on non-covalent forces. Simply by mixing an aqueous solution of fructose and curcumin in an open vessel leads to the generation of capsules with sizes ranging between 100 and 150 nm independent of the initial concentrations used. Our results demonstrate that hydrogen bonding displayed by fructose can induce the self-assembly of hydrophobic molecules such as curcumin into well-ordered structures, and serving as a simple and virtually instantaneous way of making nanoparticles from curcumin in water with the potential for template polymerization and nanocarriers.

[1] School of Chemistry, Centre for Advanced Macromolecular Design (CAMD), University of New South Wales, Sydney, NSW 2052, Australia. [2] Mark Wainwright Analytical Centre, University of New South Wales, Sydney, NSW 2052, Australia. [3] School of Chemical Engineering, University of New South Wales, Sydney, NSW 2052, Australia. [4] Australian Centre for Neutron Scattering, ANSTO, Lucas Heights, NSW 2234, Australia. [5] Present address: Department of Applied Mathematics, Research School of Physics and Engineering, Australian National University, Canberra, ANU, Australia. These authors contributed equally: Sandy Wong, Jiacheng Zhao. Correspondence and requests for materials should be addressed to M.H.S. (email: m.stenzel@unsw.edu.au)

Carbohydrates are an integral building block in nature as they are involved in numerous biological processes, which may be either structural, for example, cellulose and chitin, or functional such as cell–cell signalling[1]. Carbohydrates are commonly conjugated to proteins or lipids that play a role in stabilizing cell membranes and facilitating cellular recognition[2]. The amphiphilic character of glycolipids enables the interaction with the cell membrane resulting in structured membranes surfaces where the carbohydrates can display their multivalent character[3]. Inspired by the pivotal role these carbohydrates play in cell membranes, researchers have developed self-assembled amphiphilic glycoconjugates that can mimic the multivalency of natural carbohydrate–lectin interactions[4]. Glycoamphiphiles are usually based on covalent bonds between hydrophobic and hydrophilic segments, but non-covalent forces are also known to drive the arrangement to higher order. However, in nature as well as in the lab, amphiphiles are obtained using strong forces between the building blocks, which include ionic and covalent forces and hydrogen bonding between donor–acceptor-containing molecules.

Beyond traditional self-assembled spherical structures such as micelles, amphiphilic glycoconjugates are observed to arrange themselves into supramolecular constructs of higher order. Sugar-derived molecular gels can take on hierarchical organizations with a high aspect ratio such as fibres and ribbons, in particular when the tendency to self-assemble is superimposed with other forces such as π–π stacking[5]. It has been noted that it is the ability of sugars to participate in strong hydrogen bonding, especially when the sugar is in its cyclic form. This is among one of the key features in obtaining structures of higher order as this imparts rigidity[5] seen in the fascinating hierarchical structures obtained based on sugar amphiphiles[6–10].

To date, the formation of self-assembled structures has so far been based on a sugar surfactant where the carbohydrate is covalently bound to a hydrophobic tail[6–10]. In this communication, we report the spontaneous formation of capsules simply by adding curcumin to solutions of fructose. Curcumin, is under investigation as a potential drug against a range of diseases spanning from cancer to Alzheimer's disease[11], and displays very low water solubility (<0.1 mg/mL at pH 7)[12]. Currently, four different crystalline polymorphs are known that differ in planarity of the curcumin molecule and the position of its most prominent hydrogen bonds is present in the enol form[13]. This situation is mirrored in solution in which the enol is the dominant form. The following investigations were designed to gain insight into the nature of the nanoparticles, as well as the events that take place on a molecular level that can explain this unprecedented behaviour. Transmission electron microscope (TEM) analysis and small-angle X-ray scattering (SAXS) confirm the presence of capsules. Fundamental understanding of the molecular interactions at play was provided by theoretical studies, which reveal the thermodynamic driving force for curcumin dissolution must originate from capsule formation and fructose molecules may play a stabilizing role for structural integrity of the capsule.

## Results

**Insight to fructose–curcumin interaction with spectroscopy**. In spite of the poor solubility of curcumin in water, addition of fructose leads to a clear yellow solution that contains nanoparticulate matter (Fig. 1a). Curcumin in water at a concentration of 10 µg/mL exhibits an absorption maximum ($\lambda_{max}$) of 425 nm and a fluorescence maximum of 521 nm (Fig. 1c, d). Upon the addition of fructose solution, unequivocal changes to both the ultraviolet–visible (UV–Vis) absorbance and the fluorescence intensity can be clearly observed. In the visible band, the absorption at 425 nm decreases with increasing fructose concentration. The spectra displaying three maxima at 280 (fructose), 355 and 425 nm are similar to curcumin that has been 'clicked' covalently to sugar (Fig. 1c)[14]. According to the theoretical studies, the absorption at 425 nm can be assigned to the enol structure while the diketo isomers have an absorption maximum below 400 nm[15,16]. The enol structure of curcumin alone was confirmed using [1]H-NMR (proton nuclear magnetic resonance) studies in dimethylsulfoxide (DMSO) solvent, with the enol peak appearing at 16.51 ppm (Supplementary Figure 1). Addition of water does not lead to any isomerization, eliminating the possibility that spectral changes are the result of changes in the molecule itself. The UV–Vis spectra in Fig. 1c are devoid of an isobestic point (the intersect at 310 nm is not an isobestic point, but the result of an increasing fructose concentration). This is similar to the behaviour observed when sodium dodecylbenzenesulfonate was added to a solution of curcumin[17], the interaction of the surfactant with curcumin initially led to a decrease in the absorption at 425 nm, which is reversed when the surfactant surpasses its critical micelle concentration indicative of curcumin packed into the micelles. Although only the initial drop is observed in our case and not the subsequent band increase upon aggregation, the strong dependency of the absorption on the fructose concentration indicates the interaction of curcumin with fructose. It should be noted here that this behaviour is unlikely to be the result of altered solvent quality only, as for example, the addition of ethanol to curcumin led to an increase in the absorption at 420 nm, which was explained by the preferred arrangement of ethanol onto the enol isomers[15].

Complementing the UV–Vis results are fluorescence studies, which revealed an intense fluorescence maximum at 521 nm upon excitation at $\lambda_{ex} = 429$ nm, a wavelength that predominantly excites the enol isomer[15]. The addition of fructose led to a bathochromic shift to 560 nm, usually a sign of increased hydrophilicity and hydrogen bonding, accompanied by a decline in fluorescence intensity (Fig. 1d)[17]. Further increases in fructose translated to a slight hypsochromic shift from 555 to 551 nm and an increase in intensity indicative of potential aggregation and/or changes to microenvironment[18].

**Window of capsule formation**. It is evident that the presence of sugar affects the spectroscopic properties of curcumin, indicative of a change in its environment as is observed during its aggregation[17]. The changes in the solution can also be observed by visual inspection of the samples using higher curcumin concentrations (Supplementary Figure 2). While the mixture with low fructose content is cloudy, the presence of high concentrations of fructose results in the formation of a clear solution. To monitor the changes in particle size, dynamic light scattering (DLS) analysis was employed with fructose solutions at various concentrations, mixed with increasing amounts of curcumin (Supplementary Table 1). While no visible aggregation was observed at low curcumin concentration, the formation of nanoparticles was detected once a critical threshold of curcumin was reached (Fig. 2a). Independent of the amount of curcumin, the measured nanoparticles all had a hydrodynamic diameter of approximately 100 to 150 nm (Fig. 2a) with a slight increase in size with increasing amount of curcumin. However, further increase in the concentration of curcumin above 100 µg/mL led to larger particles and precipitation. Interestingly, independent of the fructose concentration, a critical aggregation concentration of curcumin of around 3 µg/mL, at which nanoparticles start to form, was observed. To further investigate the formation window of fructose–curcumin nanoparticles, a phase diagram was constructed. As shown in Fig. 2b, curcumin at a concentration below

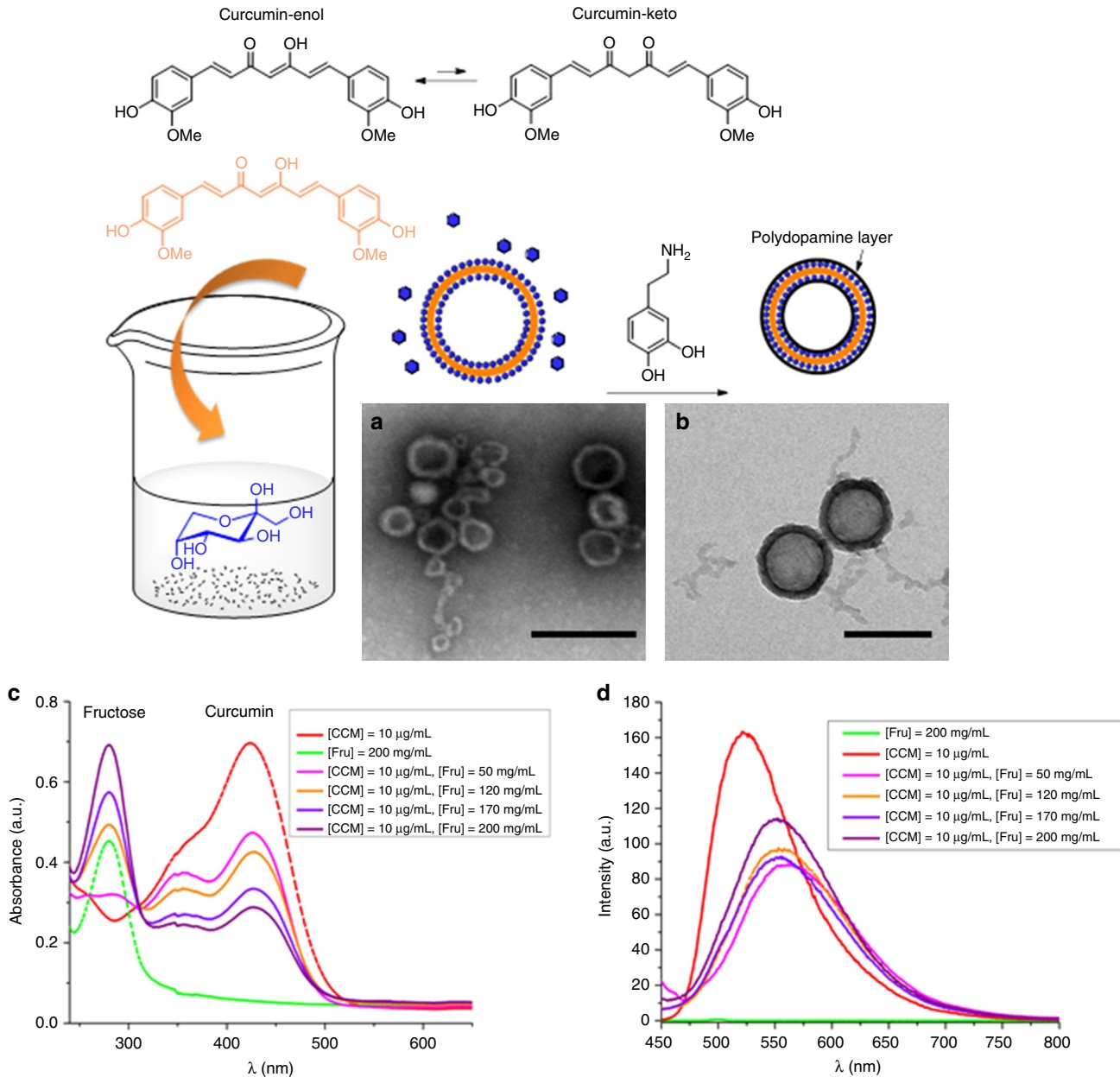

**Fig. 1** Formation of curcumin-based capsules induced by the presence of sugar and their stabilization using dopamine. The transmission electron microscope (TEM) micrographs show **a** the particles obtained using [Fructose, Fru] = 10 mg/mL, [Curcumin, CCM] = 60 μg/mL (stained with uranyl acetate, scale bar is 200 nm) and **b** the particles after the formation of a stabilizing polydopamine layer (scale bar is 200 nm), **c** UV–Vis and **d** fluorescence spectra of curcumin, fructose and curcumin/fructose mixture in water

3 μg/mL can fully dissolve in the fructose aqueous solution without the formation of any obvious aggregates. Once the curcumin concentration reached the critical threshold of 3 μg/mL, fructose and curcumin start to aggregate to form nanoparticles. The formation window spans from a critical curcumin concentration of around 3 to 80 μg/mL. Within the window of nanoparticle formation, the observed hydrodynamic diameters were constant and not significantly dependent on the fructose or curcumin concentrations. However, high fructose concentration tends to limit assembly of curcumin by inhibiting its dispersion and results in its precipitation (Supplementary Figure 2). Although minute amounts of DMSO were used in the process, the outcomes are independent from the type of solvent as other solvents such as ethanol, tetrahydrofuran or acetone result in similar outcomes (Supplementary Tables 2 and 3).

**Morphology of nanocapsules**. The morphology of the capsules was further confirmed by TEM. Spherical nanoparticles reminiscent of hollow capsules were observed with sizes similar[19] to those measured using DLS (Fig. 1 and Supplementary Figure 3). Within the window of nanoparticle formation as highlighted in Fig. 2b, capsules can be observed, while higher fructose or curcumin concentrations favour the formation of ill-defined aggregates based on solid nanoparticles (Supplementary Figure 4). To confirm the true size of these capsules and eliminate the drying effect in conventional TEM, cryo-TEM was performed to observe the particles in their frozen-hydrated state. No aggregation was observed, and the capsules were found to have around a 100 nm hydrodynamic diameter (Fig. 2c); in excellent agreement with size determination by DLS (Fig. 2a). The capsules are characterized by a transparent fructose layer on the surface of an electron-rich

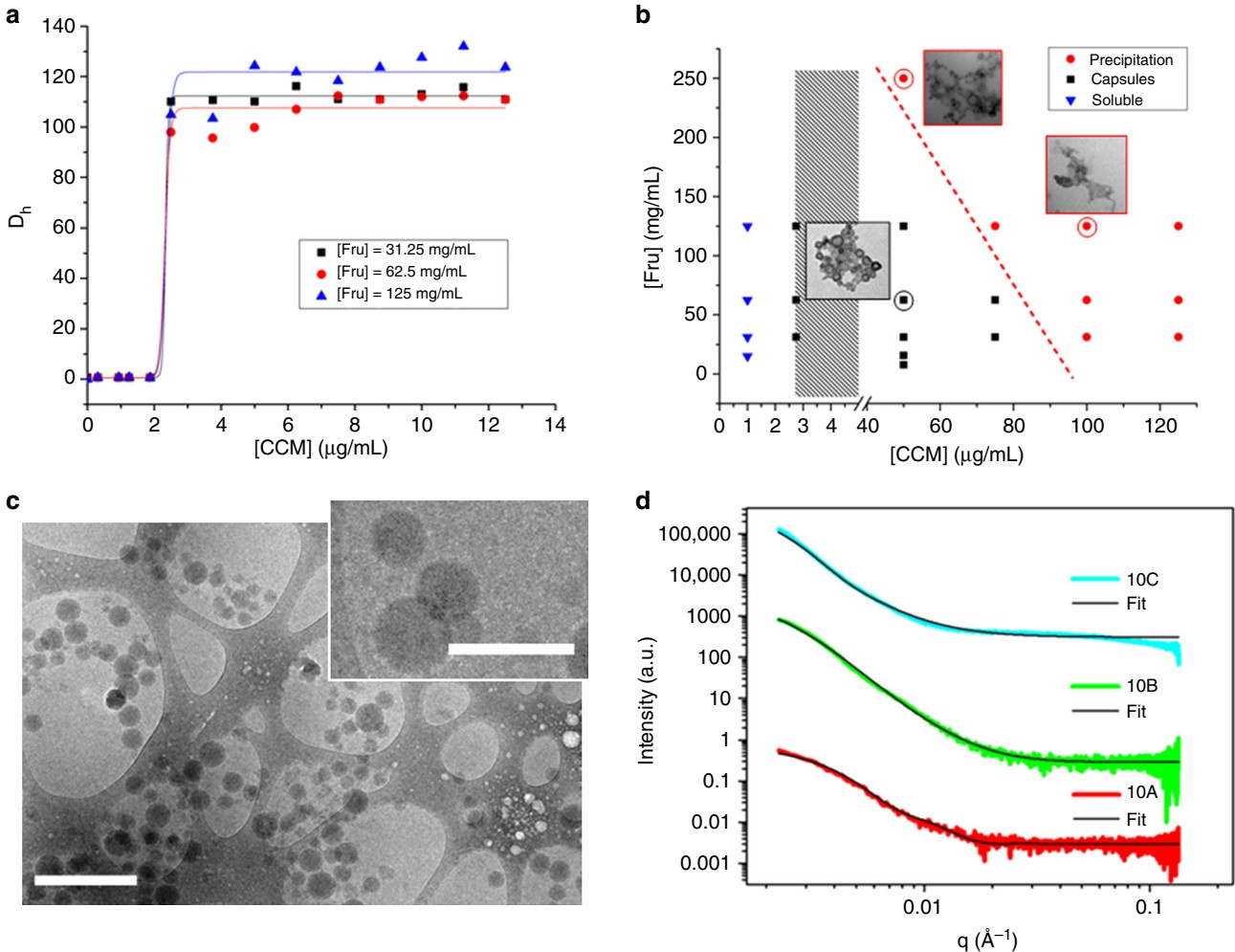

**Fig. 2** Analysis of size and structure of the nanocapsules. **a** Curcumin concentration dependence of the nanoparticle size, **b** phase diagram of fructose–curcumin nanoparticles; grey bar denotes area of aggregates and/or capsules, **c** cryo-transmission electron microscopy (TEM) of the nanoparticles prepared with [Fructose, Fru] = 10 mg/mL and [Curcumin, CCM] = 60μg/mL (scale bar is 500 nm, scale bar in inset is 200 nm) and **d** small-angle X-ray scattering (SAXS) measurements with a fixed fructose concentration (10 mg/mL) and varying curcumin concentration (where A = 20 μg/mL, B = 60 μg/mL and C = 80 μg/mL) and their fitting

curcumin packed layer, that encompass the hollow structure. Interestingly, the absence of a clear ring-like outline in cryo-TEM (Fig. 2c) suggest a fuzzy interface constructed by small molecules, rather than a well-defined boundary typically seen with self-assembled lipid and polymer-based vesicles. This phenomena has been observed before, where loose membrane structures led to the disappearance of an outside ring[20].

To gain more insights into the surface, size properties and shape of the nanoparticle, SAXS analysis was carried out (Fig. 2d). All SAXS data were fitted to a core-shell model[21] and summarized in Table 1. The scattering length density of the solvent ($SLD_{solvent}$) was fitted with a similar value to the SLD of the core, which is consistent with a solvent-filled hollow core (Supplementary Tables 4 and 5). Increasing amounts of curcumin at a fixed fructose concentration led to a change in $SLD_{solvent}$ from $1.02 \times 10^{-5}$ to $9.59 \times 10^{-6}$. This may be caused by an increase in the number of nanoparticles and a decrease in unbound fructose in solution. The 10C fitting is rough as the exact fructose concentration in the vesicle cannot be confirmed due to low concentrations. However, the fitting shows changes when increasing the amount of curcumin contributes to the nanoparticle. Interestingly, when all measured samples were left

standing for more than 24 h, objects with a slope of $n = -4$ for log(Intensity) vs. log($q$) (Supplementary Figure 5 to 7) appeared. This is indicative of larger objects with zero curvature on the length scales considered by the scattering vector according to the

**Table 1 SAXS data for samples at fixed [Fru] = 10 mg/mL (10) and varying [CCM], where A = 20 μg/mL, B = 60 μg/mL and C = 80 μg/mL, and their fitting parameters**

| Sample | 10A | 10B | 10C |
|---|---|---|---|
| Fructose (mg/mL) | 10 | 10 | 10 |
| Curcumin (μg/mL) | 20 | 60 | 80 |
| Core radius (nm) | 18.3 ± 1.6 | 36.6 ± 1.0 | 59.8 ± 2.0 |
| $SLD_{core}$ ($\times 10^{-5}$ Å$^{-2}$) | 1.02 ± 0.074 | 1.12 ± 0.002 | 1.03 ± 0.002 |
| Shell thickness (nm) | 29.9 ± 0.4 | 8.9 ± 0.8 | 8.0 ± 1.0 |
| Diameter (nm) | 96.4 | 90.2 | 135.6 |
| $SLD_{shell}$ ($\times 10^{-5}$ Å$^{-2}$) | 1.11 ± 0.002 | 1.16 ± 0.0014 | 1.16 ± 0.004 |
| $SLD_{solvent}$ ($\times 10^{-5}$ Å$^{-2}$) | 1.02 ± 0.0074 | 0.979 ± 0.0024 | 0.959 ± 0.003 |

*SAXS small-angle X-ray scattering, SLD scattering length density, Fru fructose, CCM curcumin*

Porod Law. The results were found to be independent of the fructose and curcumin concentrations, as all measured particles followed the same relationship. The intercept of the intensity plot at $q$-$>0$ for $I(q) * q^4$ vs. $q^4$ was found to decrease with increasing amount of curcumin (Supplementary Figure 7) reflecting an increasing number of particles in solution, which is consistent with the observed decrease of the $SLD_{solvent}$. Importantly, the SAXS measurements demonstrate that the objects are similar in size to the ones measured by DLS (Fig. 2a). However, the increase in shell thickness at lower curcumin concentration suggests a shell of lower density and corresponds to lower $SLD_{shell}$ value.

SAXS and DLS were used to gather information on the nature of self-assembled capsules with similar size independent of curcumin and fructose. It seems that increasing the amount of curcumin predominantly changes the number of particles, instead of their physical properties (Supplementary Figure 8). The size of the capsules is governed by the membrane thickness and the surface curvature, which in turn is affected by the nature of amphiphiles. The formation of capsules is due to the bending of the bilayered membranes, with the diameter of the resulting capsules being influenced by membrane elasticity of the amphiphilic molecule and membrane thickness[22]. This suggests that independent of their concentrations, a unique amphiphilic construct is present in a particular phase window, and is not influenced by the concentration of either compounds, but by the intermolecular forces specific to curcumin and fructose.

**Structural stability based on intermolecular forces**. It is evident that the interaction of the sugar and curcumin causes the formation of molecular self-assembled structure without any apparent amphiphilic molecules being present. Supramolecular assemblies based on forces such as ionic bonds, $\pi$–$\pi$ interactions and hydrogen bonding are well known. The concept of forming vesicles based on non-traditional amphiphiles has already been discussed in an early review by Antonietti and Förster[23], who highlighted the use of ionic and complementarity interactions in polymers as a means of generating vesicles[24,25]. Beyond polymeric systems, low-molecular-weight compounds based on guanosine are known to arrange themselves into higher-order structures simply through the strong donor–acceptor interactions between carbonyl groups and protonated amines[18]. Vesicles based on the self-assembly of guanosine[26,27], but also other strong hydrogen-bonding motifs, have been mentioned previously[28,29]. Complementing this exploitation of hydrogen bonding are amphiphiles that are generated in situ by ionic interactions with small molecules bearing opposite charges[30–34].

However, sugars used here, such as fructose and curcumin, are devoid of any electrostatic charge interactions. Also, strongly matching hydrogen bonding between carbonyl groups and protonated amines, typical for DNA inspired structures and the driving force for directed self-assembly[18,35] are also absent in this study. It should be noted that in nature a variety of biomacromolecules, including enzymes, proteins and DNA are constructed via well-controlled intermolecular and/or intramolecular interactions in water. For this reason, it seems that the weaker hydrogen bonding between the fructose and curcumin hydroxyl groups are sufficient to drive the self-assembly in water. Furthermore, factors that affect the hydrogen bonding such as temperature (Supplementary Figure 9) as well as pH (Supplementary Figure 10) play a role. Increasing the temperature resulted in a slight shift in fluorescence maximum toward free curcumin, which suggests reasonably good stability at higher temperatures before signs of disassembly at 45 °C well above

room temperature. In contrast, lowering the pH resulted in disassembly and precipitation, suggesting the perturbation of the delicate intramolecular and intermolecular hydrogen bonding through protonation of the hydroxyl groups. Osmotic pressure along with gradual removal of fructose caused preassembled nanoparticle aggregation within 3 h, slight disruption of capsule morphology and steady decline in derived count rate (kcps) can be observed over time (Supplementary Figure 11). Thus, stressing the importance of carbohydrate interaction.

**Theoretical studies**. To better understand the origin of the enhanced solubility of curcumin molecules in fructose–water solution, molecular dynamics (MD) simulations were carried out to compare the free energy of solvation of curcumin in water vs. fructose solution. In the former, the simulation consists of a single curcumin molecule immersed in a 5 nm × 5 nm × 5 nm periodic box of TIP3P water molecules. MD simulations in conjunction with the weighted histogram analysis method (see Supplementary Figure 12 and Table 6 for details) were employed to determine the potential of mean force (PMF) along the reaction coordinate defined to be the distance between centres of mass of the solute and the solvent box. This is shown in Supplementary Figure 12b along with the corresponding PMF for curcumin immersed in the same periodic box containing water and 200 fructose molecules to mimic the fructose solution. Interestingly, the solvation free energies (obtained as the difference in PMF at the end points of the reaction coordinate) are very similar, ca. 15.7 and 16.8 kcal/mol in water and fructose solution, respectively. According to the thermodynamic cycle presented (Supplementary Figure 12), the modest enhancement in solvation of curcumin in fructose solution implies that the thermodynamic driving force for curcumin dissolution must originate from capsule formation, the $\Delta G$(capsule).

The self-assembly process is estimated to occur on seconds timescale, which is difficult to probe using conventional MD simulations. As such, a planar model of the capsule (8 nm × 8 nm × 12 nm) composed of 200 curcumin molecules, 700 fructose and 12,000 water molecules was constructed using PACKMOL[36] to examine the structural integrity of the fructose–curcumin–fructose membrane using MD simulations. Figure 3a shows the starting structure of the model, and a snapshot after equilibration (Fig. 3b). In the initial configuration, the curcumin molecules were packed in random orientations. Comparison of the two snapshots revealed that fructose molecules readily diffuse into bulk water; however, the MD simulations indicate that there is a fructose concentration profile that decreases as the distance from the curcumin layer increases (Fig. 3c). This is in line with SAXS findings described herein.

To probe the effect of fructose on the stability of the curcumin layer, we have also performed a set of simulations using an identical periodic cell, but with fructose molecules replaced by water molecules. Within a 100 ns trajectory, the curcumin layer remains stable under zero lateral tension; however, the projected area per curcumin appears to be smaller compared to the simulations where fructose is present (21.9 Å$^2$/curcumin compared to 24.0Å$^2$/curcumin after 100 ns). This suggests that there is greater tension between curcumin and water, and that fructose helps to mediate the interactions between these molecules. From Fig. 3d, it is evident that these simulations have not equilibrated after 100 ns—the area per curcumin is still shrinking. Presumably, over a longer time scale, the curcumin layer in the simulations without fructose would eventually collapse; however, it is impractical to probe this using conventional MD simulations. We have also performed MD simulations where fructose

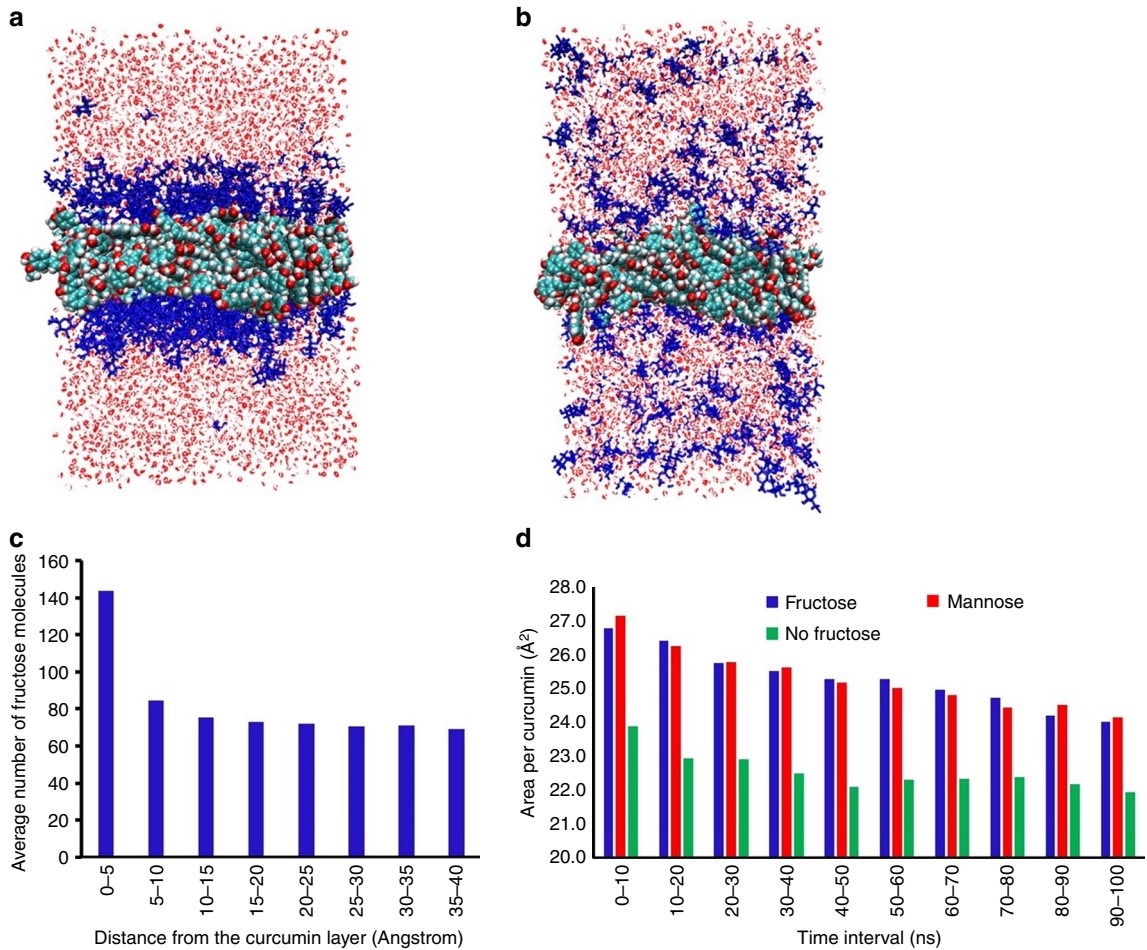

**Fig. 3** Snapshot from the molecular dynamics (MD) trajectory of a fructose–curcumin–fructose membrane model **a** prior to equilibration and **b** after equilibration. **c** The MD simulations indicate that there is a concentration gradient of fructose molecules that decreases with increasing distance from the curcumin layer. **d** The average area per curcumin at various time intervals along a 100 ns trajectory for simulations with and without fructose, and with mannose

molecules are replaced with mannose, which did not result in hollow particles (vide infra). As shown in Fig. 3d, the area per curcumin ($24.2 Å^2$) is comparable to the simulations with fructose. This suggests that the conventional fixed charge force fields may not be accurate enough to resolve the level of subtlety between fructose and mannose.

While the timescale of these simulations (100 ns) are short compared to experiment, they suggest that fructose molecules may play a stabilizing role in mediating the interaction between curcumin molecules and water that may be important for structural integrity of the capsule.

**Importance of maximum hydrogen bonding and other sugars**. The formation of hollow nanostructures instead of solid spheres may be ascribed to the specific association of curcumin and the sugar. The hydrogen bonding between the phenolic hydroxyl groups and the α,β-unsaturated carbonyl groups of curcumin are dominant[13]. To test this hypothesis, divanillin, which has a similar structure to curcumin but devoid of the α,β-unsaturated carbonyl groups, was used for self-assembly. Interestingly, only solid spheres were observed (Supplementary Figure 14), suggesting the important role of functional groups between two aromatic rings in the structure of curcumin. Stripping curcumin down even further, vanillin (precursor of the symmetrical dimer

divanillin) led to no self-assembled nanoparticles in the presence of fructose. The count rate of scattered light detected by DLS is less than 20 kcps for the measurement, which further indicated the pivotal role of the α,β-unsaturated carbonyl and phenolic alcohol interactions between curcumin molecules.

The ability to form hollow particles in aqueous solution with or without carbohydrates can be tested to highlight the integral role of hydrogen bonding in self-assembly. Without sugar, a similar nanoparticle system would not exist merely in water (Supplementary Figure 15) and render curcumin insoluble as large precipitates. Other sugars like glucose and mannose were also tested in the formation of capsules with curcumin (Supplementary Figure 16). Glucose, as an isomer of fructose, formed identical hollow nanoparticles with curcumin under the same preparative conditions. In contrast, hollow nanostructures were not observed with mannose and galactose, and instead significant precipitation and uncontrolled aggregation was evident in TEM. This difference in behaviour can be attributed to the orientation of individual hydroxyl groups on the sugars that were investigated. The two axial -OH groups in both galactose and mannose can participate in intramolecular hydrogen bonding networks depending on the relative configuration of other -OH groups (Supplementary Figure 17). A more nonpolar-like hydration shell could be expected to surround its hydroxyl groups, which can limit other intermolecular hydrogen bonding

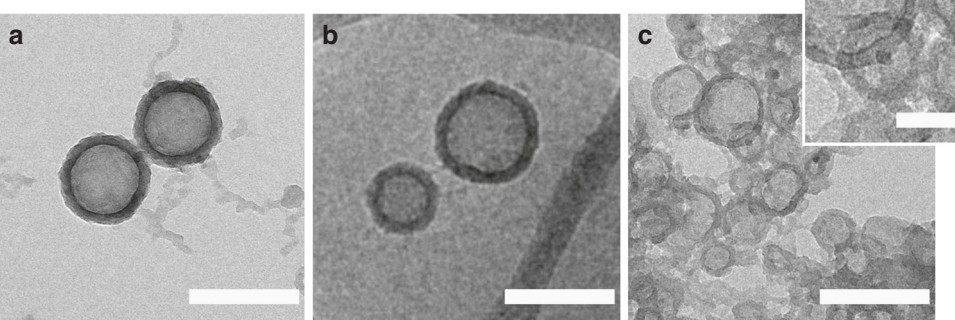

**Fig. 4** Microscopy of PDA-coated nanocapsules. **a** Transmission electron microscopy (TEM) and **b** cryo-TEM micrographs of polydopamine–fructose-curcumin (PDA–Fru–CCM) capsules before dissolution in organic solvent (methanol) and **c** TEM micrograph after dissolution in organic solvent and subsequent removal of curcumin membrane from polydopamine shells. Scale bars in **a**–**c** are 200 nm, scale bar inset in **c** is 50 nm

interactions[37]. The ordering of hydrogen bonding strength of carbohydrate hydroxyl groups is generally axial–axial > axial–equatorial > equatorial–equatorial[38]. This complements the fact that fructose in the current study can be predominantly found as β-D-fructopyranose, which has predominantly equatorial hydroxyl groups (Supplementary Figure 18). The hydroxyl groups of glucose exist in equatorial positions as the lowest energy conformation and the three hydroxyls of the fructose ring form stronger hydrogen bonding with a solvent than its hydroxyls explained by MD simulations[39]. It should be noted that the structural orientation of the carbohydrate used should not be ignored in maximizing hydrogen bonding capacity with -OH functionality of curcumin.

**Template polymerization.** As the self-assembled structure of these small molecules is fragile, the capsules were captured and preserved by coating with polydopamine. The fructose–curcumin hollow nanoparticles were prepared by dropping a solution of curcumin in DMSO into a solution of aqueous fructose, followed by the addition of dopamine hydrochloride and tris (tris(hydroxymethyl)aminomethane) base solution to trigger the oxidative polymerization of the dopamine as the capsules formed in situ (Fig. 1). To further confirm that the hollow nanostructure was unaffected during the self-polymerization of dopamine, TEM was employed to investigate the morphologies of the polydopamine-coated nanoparticles. As shown in Fig. 1 and Supplementary Figure 19, hollow nanoparticles with a membrane structure were observed after surface coating. As anticipated, after coating, the thickness of the membrane structure increased to 25 nm, suggesting the successful polymerization of dopamine on the surface of fructose–curcumin nanoparticles bearing a rough topography and revealed a central cavity upon drying (Supplementary Figure 20). The resulting polydopamine hollow spheres could be collected by filtration or centrifugation. Isolation of the nanoparticles enabled analysis of the remaining clear solution, which was devoid of curcumin. Fluorescence analysis, which typically displays maximum emission between 488 to 527 nm, confirmed that all curcumin was involved in the hollow particle formation process.

Deformations from collapsed polydopamine fructose–curcumin capsules show folds and creases that are typical of capsules that would be unlikely possible if it were solid spheres to collapse (Supplementary Figure 20). Location of the fructose–curcumin membrane was determined by removal upon organic solvent dissolution (i.e., methanol) and dialysis back to aqueous solution (Fig. 4). The remainder are polydopamine shells that likely occupy

both inside and outside the fructose–curcumin membrane layer. Due to the concentration gradient, small molecules such as dopamine can diffuse through the porous structure and subsequently adhere and polymerise on the inner side of the membrane. A similar case can be observed with dye (Fig. 1a and Supplementary Figure 3), diffusion into the hollow cavity indicates a relatively porous membrane that complements SAXS finding of an ill-defined boundary between hydrophilic and hydrophobic moieties when prepared fresh. In contrast to polymer vesicles that are promising candidates for drug/gene delivery, our study does not require traditional self-assembly of block copolymers that involve tedious synthetic steps. These supramolecular constructs can be adopted in template polymerization, in which the formation of the template is quick and easily removed after polymerization while maintaining the structural integrity of polymer formed. This system is biocompatible, easy to prepare, require little organic solvents and present possible routes to be adopted as nanocarriers and/or nanoreactors.

## Discussion

We have shown here that in aqueous solutions, sugars can display sufficiently powerful hydrogen bonding with curcumin, to promote their co-self-assembly into capsules. Although non-covalent amphiphiles are known, we believe that this is the first report that describes the formation of self-assembled structures that are not based on ionic or the stronger nitrogen-based hydrogen bonding. It highlights the probably underappreciated strength of molecular interactions based on sugars. The only prerequisite is an appropriate arrangement between sugar and substrate, here curcumin allows the close proximity between the hydrogen donor and acceptor groups. Hydrogen bonding is an integral part of water and its unusual chemical and physical properties are often explained using water clusters[40]. This report demonstrates that it is feasible to consider sugar not only as a sweet and bioactive group, but additionally a molecule whose strong hydrogen bonding can play an integral role in self-assembly.

## Methods

**Preparation of fructose–curcumin hollow particles.** Taking [Fructose] = 31.25 mg/mL as an example: fructose (31.25 mg) was dissolved in milliQ water (1 mL). To this solution, curcumin in DMSO (20 μL, 2.5 mg/mL) was added dropwise with gentle stirring (120 rpm) resulting in a solution with a final curcumin concentration of 50 μg/mL. The procedure was repeated with different fructose concentrations (62.5 and 125 mg/mL) and with different volumes of curcumin–DMSO solution.

**Preparation of polydopamine–fructose–curcumin hollow nanoparticles**. A typical procedure to prepare polydopamine–fructose-curcumin nanoparticles is described as follows: to a solution of fructose (62.5 mg) in milliQ water (2 mL) at room temperature, with stirring (120 rpm), curcumin/DMSO solution (40 μL, 2.5 mg/mL) was added dropwise. The fructose solution turned from colourless to yellow. After stirring for an additional 5 min, dopamine hydrochloride aqueous solution (10 μL, 10 mg/mL) and 10 μL Tris base aqueous solution (6 mg/mL) were added to the curcumin/fructose solution sequentially. Then, the vial was covered with foil and left for polymerization. After 48 h, the reaction solution was dialyzed against milliQ water before characterization.

## Data availability

The datasets generated during and/or analysed during the current study are available from the corresponding author on reasonable request.

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

## Acknowledgements

We would like to acknowledge the Mark Wainwright Analytical Centre at the University of New South Wales for providing access to instruments at the Electron Microscope Unit and the NMR Facility. We would like to thank the Australian Synchrotron for access to the SAXS beamtime. We also like to thank Ms. Shanshan (Claudia) Qi and Mr. Guannan Wang for their support in DLS and SEM measurements, respectively. S.W. is grateful for UNSW PhD scholarship. J.H. acknowledges support from the Australian Research Council (DE160100807) and supercomputer resources from the NCI, Pawsey Super-computing Centre and Intersect Australian Ltd. Finally, M.H.S. and C.J.G. would like to thank the Australian Research Council (ARC DP 160101172) for funding.

## Author contributions

S.W. performed the experiments and analyzed the data, J.Z. discovered the initial find-ings. S.W. and R.P.K performed TEM measurements, C.K.W. and R.P.K. conducted cryo-TEM measurements. J.M.H. helped with the NMR analysis. C.C. and C.J.G. carried

out the scattering experiments and data analysis. S.S. and S.D.L. performed initial computational studies, theoretical studies shown in this report was done by J.H. M.H.S. coordinated all activities and discussions and wrote the first draft of the manuscript. All authors discussed the results and commented on the manuscript.

## Additional information

**Competing interests:** The authors declare no competing interests.

