## [Peer Review File · Nature Communications]

Reviewers' comments:

Reviewer #1 (Remarks to the Author):

The authors claim how the hydrogen bond displayed by fructose can induce self-assembly of hydrophobic molecules such as curcumin into well-ordered structures and serve as a simple and virtually instantaneous way of producing nanoparticles from curcumin in water. They provided more than one insight describing the results of SAXS, DLS and TEM, as well as MD simulation and a large number of well organized support informations. In my opinion, statistical analysis is appropriate and the author should be able to reproduce the results because they provided a high level of details and images of the experimental procedure performed. To the best of my knowledge, this is the first paper showing in aqueous solutions, the power of sugars to bind with curcumin by promoting their self-assembly into vesicles. There are some published articles that report the use of self-assembled phytosterol-fructose-chitosan nanoparticles and fructose-coated nanoparticles as carriers of an anticancer drug, but the possibility of using a simple molecule for the organization of a highly insoluble compound such as curcumin in my opinion is new. Curcumin is a natural polyphenol that can bind the beta-amyloid peptide, which is related to Alzheimer's disease (AD), and modify its self-assembly pathway. Therefore, the results are interesting because they are focused in the field of emerging nanotechnologies based on targeted nanomedicines that are non-invasive (fructose is a biocompatible compound) transport of highly potent and specific drugs (or natural compounds such as curcumin) through the blood barrier -encephalic. Then, the results might be promising for a new therapy for AD even if carefulness is always required.

Reviewer #2 (Remarks to the Author):

The contribution from Jiacheng Zhao et al. aims at demonstrating that hydrogen bonding displayed by saccharide molecules such as fructose can induce the self-assembly of hydrophobic molecules such as curcumin into well-ordered structures, and serving as a simple and virtually instantaneous way of making nanoparticles. The experiments proposed evidenced the spontaneous formation of stable nanoparticles when mixing fructose and curcumin in water, as evidenced by light scattering, TEM and UV and fluorescence measurements. Even if glycolipids or sugar-based surfactants are known to form self-assembled structures, such in situ and spontaneous interaction from saccharide and hydrophobic molecules to afford stable nanoparticles is original and present and interesting therapeutic potential. The paper is generally nicely written and illustrated. To my opinion, the paper may present the potential to be published in Nature Communications, but not in the present form. Indeed, some major experimental and interpretation discrepancies need to be corrected and discussed in more details, especially concerning the mechanism of stabilization and the nanoparticle morphology.

The formation of nanoparticles is rather obvious and clear. However, the vesicle morphology is not clear and not supported by the experimental data proposed. TEM on Figure 1A showed darker external layer that can be due to staining effect. Figure 1B evidenced a kind of membrane, but that may be due to the polydopamine external layer around the particle. There is no real evidence of vesicle structure here. Figure 2 E, showing cryo-TEM, does not bring any better evidence. Even if it is claimed by the authors that a thin bilayer is formed, it should be observed (especially if it is supposed to be in between 16 and 25nm as depicted at the end of the manuscript, which is not thin at all). Bilayers from liposomes, that are very thin (3-5nm) can clearly be observed by cryo-TEM. Here, the objects look really homogeneous, with the same density, resembling a particle, not a vesicle. In addition, SAXS data proposed on Figure 2D are also not very meaningful. The sharp interface between the nanoparticle's surface and solvent is evidence from the q^{-4} slope. A thin vesicle membrane should present a q^{-2} slope in this q -range, which is not the case here. Altogether, I have strong doubt about the reality of such vesicle formation. Better cryo-TEM images should be provided or other evidences based on scattering (SANS, SLS,...) to fully evidence the formation of vesicles, if any.

The other very strong weakness is about the mechanism of formation and stabilization of the

nanoparticles. Previous contributions about the formation of vesicles involving strong H-bonding are referred. However, in these systems (ref. 24, 25), H-bonding was occurring in organic solvent. Here, the formation of H-bonding between curcumin and sugar in water is not obvious. It is well known that water is not a favorable solvent to stabilize such interactions (even worse if polar solvents such as DMSO are also present). Temperature or pH variations are not sufficient to prove the presence of H-bonding. Why not adding urea for example? In addition, sugars are also known to present a hydrophobic character (see for instance: Hydrophobic nature of sugars as evidenced by their differential affinity for polystyrene gel in aqueous media. Janado, M. & Yano, Y. J Solution Chem (1985) 14: 891). The sugars could also stabilize the interface of nano-clusters of curcumin by absorption at the interface, due to their hydrophobic character.

I have a major concern about the DLS measurements. It is not described in the experimental part and ESI how the solvent viscosity is corrected to determine accurately the Rh. Indeed, when sucrose and curcumin (including organic solvent such as DMSO) concentrations are varied, how the changes in dn/dc and viscosity are considered? The size of the nanoparticles is presented to be constant, but is it really the case?

Reviewer #3 (Remarks to the Author):

This work by Zhao and coworkers characterizes small hollow spheres (vesicles) formed by the relatively non-polar curcumin molecule in fructose solution. The proposed importance of the observation is that it indicates a new type of self-assembled structure — formed of two components that are not covalently linked. The vesicles have diameters of approximately 100 nm and surface thickness of 16 nm. This is characterized by electron microscopy (Figure 1) and dynamic light scattering (Figure 2). Spectroscopy indicates that the electronic environment around the curcumin and fructose changes substantially as part of the vesicle structure (Figure 1) implying they are interacting.

I find the observation of these structures to be interesting and certainly to defy my expectations. My expectation is that curcumin would always form solid particulate assemblies to minimize its apolar surface exposure. In lipid bilayers vesicles this route is not possible because the polar piece is covalently bonded to lipid. Fructose is rather soluble in water. So is something strong happening at the fructose-curcumin interface (analogous to the strong bonds in amphiphiles)? Or, as I prefer below, are these vesicles actually very small bubbles that are perhaps kinetically stabilized by fructose?

The paper does not answer this important question. Molecular dynamics simulations were employed to investigate the chemical mechanism of the self assembly. Potentials of mean force (PMFs) between curcumin and fructose are the key of their simulation "experiment". These simulations showed that curcumin was more attracted to itself (3 kcal) than to fructose (1 kcal). Kcal is the reasonable energy scale for these interactions. A "control" for fructose-fructose would have completed this picture. So, shouldn't curcumin simply aggregate according to our expectation? Experiments with other sugars indicate that vesicles aren't formed. Would PMFs between curcumin and these other sugars indicate a difference? Unfortunately this level of subtlety is likely below the ability of molecular dynamics forcefields to help with. In summary, the PMFs, at least superficially, do not indicate that the surface should be stable. Rather, they indicate that it should collapse because the curcumin-curcumin interaction is so strong.

Another confusing point was that the thickness of the surface by electron microscopy was 16 nm. This is ca. 10 times thicker than well above the thickness of curcumin. There must be many layers of curcumin in the surface. Is fructose incorporated in the surface center? Why is the surface so thick?

To me this points to metastability — that water is trapped inside and that the assembly would prefer to collapse. that is, it is a bubble, not a vesicle. Vesicles do not get popped. If it is a bubble, it is certainly interesting that they are so small and so uniform. It is also appears clear that fructose is acting as a surfactant between curcumin and water, lowering the tension and thus lowering the energy of the bubble.

A key question is whether the surface are stable under zero lateral tension, like that of a lipid bilayer. Here tension may be applied by the Young-Laplace law.

Because the assembly is only stable in fructose, the spheres will naturally disintegrate when transferred outside of fructose. This makes there application extremely limited: limited to solutions of fructose. The authors write: "As the self-assembled structure of these small molecules is fragile, the vesicles were captured and preserved by coating with polydopamine."

Experiments that would substantially alter interest in this work would address the stability of the spheres themselves. Under size exclusion chromatography to remove the fructose solution do the particles collapse? Can they withstand any level of sonication? Can they withstand osmotic stress? Can you make a Langmuir-Blodgett film of this stuff?

The authors cannot simulate the 100 nm diameter particle because the simulation is too large. Can they simulate a planar bilayer of the "material" much as planar lipid bilayers are simulated to represent large vesicles? If the surface is stable at zero tension this would be very interesting. If it collapses (ie, it thickens uncontrollably to reduce its projected area) it may indicate extreme metastability. The tension of the curcumin-water surface with and without fructose could be compared by simulating planar sheets at fixed projected area and reporting the tension as is commonly output in molecular dynamics simulations. Can it be reduced to zero? Current forcefields are not perfectly reliable for the surface tension between polar and apolar compounds but the answers might provide some hints as to what the fructose is doing.

I do not have any concerns about a lack of information impeding reproducibility. The recipe for these is simple.

Minor issues:

Their presentation of the small angle x-ray scattering (SAXS) data, intended to show a sharp surface of the particles was very difficult to parse. Electron microscopy images of the samples visually indicate that the surfaces are not sharply defined, that in fact the thickness varies fairly dramatically. The SAXS data appear to rebut this. Figure S5 appears to show perfect agreement to Porod's law. Figure S6, presumably similar data plotted differently, appears to show much noisier data than S5.

For some sentences the english was so poor that the meaning was not clear. "It has been noted that it is the ability of sugars to participate in strong hydrogen bonding, especially when the sugar is in its cyclic form, that is one of the key features in obtaining structures of higher order as this imparts rigidity, such as the fascinating hierarchical structures based on sugar amphiphiles" and "Currently, four different crystalline polymorphs are known, which differ in the planarity of the curcumin molecule and as a consequence, in the position of its most prominent hydrogen bonds, is present in the enol form".

Prof Martina Stenzel
CAMD
School of Chemistry
University of New South Wales
UNSW Sydney NSW 2052
Australia
Phone: +61-2-9385 4656
<http://www.camd.unsw.edu.au>
M.Stenzel@unsw.edu.au

Sydney, 1/8/2018

“**Just add sugar – carbohydrate induced self-assembly of curcumin**” by *Sandy Wong,[#] Jiacheng Zhao,[#] Cheng Cao, Chin Ken Wong, Rhiannon P. Kuchel, Sergio De Luca, James M. Hook, Christopher J. Garvey, Sean Smith, Junming Ho, Martina H. Stenzel*

Dear reviewers

We would like to thank the reviewers for their interest in our paper and their insightful comments that helped us to improve our manuscript. All reviewers found our concept interesting, but they also shared their thoughts with us on how to interpret results differently.

We hope that the revision can shine more light on this very unusual interaction between curcumin and sugar.

Kind regards

Martina Stenzel and co-authors

Black = The reviewer comments

Red = Response to reviewers

Blue = The Changes in the MS

Responses to Editorial Office and reviewers:

Reviewer 1:

The recommendation of **Reviewer 1** is “*The authors claim how the hydrogen bond displayed by fructose can induce self-assembly of hydrophobic molecules such as curcumin into well-ordered structures and serve as a simple and virtually instantaneous way of producing nanoparticles from curcumin in water. They provided more than one insight describing the results of SAXS, DLS and TEM, as well as MD simulation and a large number of well-*

organized support informations. In my opinion, statistical analysis is appropriate and the author should be able to reproduce the results because they provided a high level of details and images of the experimental procedure performed. To the best of my knowledge, this is the first paper showing in aqueous solutions, the power of sugars to bind with curcumin by promoting their self-assembly into vesicles. There are some published articles that report the use of self-assembled phytosterol-fructose-chitosan nanoparticles and fructose-coated nanoparticles as carriers of an anticancer drug, but the possibility of using a simple molecule for the organization of a highly insoluble compound such as curcumin in my opinion is new. Curcumin is a natural polyphenol that can bind the beta-amyloid peptide, which is related to Alzheimer's disease (AD), and modify its self-assembly pathway. Therefore, the results are interesting because they are focused in the field of emerging nanotechnologies based on targeted nanomedicines that are non-invasive (fructose is a biocompatible compound) transport of highly potent and specific drugs (or natural compounds such as curcumin) through the blood barrier -encephalic. Then, the results might be promising for a new therapy for AD even if carefulness is always required.”;

We thank the kind reviewer for their interest in our work and still find this paper appropriate to be published in Nature Communications along with recent revisions.

Reviewer 2:

The recommendation of **Reviewer 2** is “The contribution from Jiacheng Zhao et al. aims at demonstrating that hydrogen bonding displayed by saccharide molecules such as fructose can induce the self-assembly of hydrophobic molecules such as curcumin into well-ordered structures, and serving as a simple and virtually instantaneous way of making nanoparticles. The experiments proposed evidenced the spontaneous formation of stable nanoparticles when mixing fructose and curcumin in water, as evidenced by light scattering, TEM and UV and fluorescence measurements. Even if glycolipids or sugar-based surfactants are known to form self-assembled structures, such in situ and spontaneous interaction from saccharide and hydrophobic molecules to afford stable nanoparticles is original and present and interesting therapeutic potential. The paper is generally nicely written and illustrated. To my opinion, the paper may present the potential to be published in Nature Communications, but not in the present form. Indeed, some major experimental and interpretation discrepancies need to be corrected and discussed in more details, especially concerning the mechanism of stabilization and the nanoparticle morphology.”.

Response and revisions:

1/ “The formation of nanoparticles is rather obvious and clear. However, the **vesicle morphology** is not clear and not supported by the experimental data proposed. TEM on Figure 1A showed darker external layer that can be due to staining effect. Figure 1B evidenced a kind of membrane, but that may be due to the polydopamine external layer around the particle. There is no real evidence of vesicle structure here. Figure 2 E, showing cryo-TEM, does not bring any better evidence. Even if it is claimed by the authors that a thin bilayer is formed, it should be observed (especially if it is supposed to be in between 16 and 25nm as depicted at the end of the manuscript, which is not thin at all). Bilayers from liposomes that are very thin (3-5nm) can clearly be observed by cryo-TEM. Here, the objects look really homogeneous, with the same density, resembling a particle, not a vesicle. In addition, SAXS data proposed on Figure 2D are also not very meaningful. The sharp interface between the nanoparticle’s surface and solvent is evidence from the q^{-4} slope. A thin

vesicle membrane should present a q^{-2} slope in this q -range, which is not the case here. Altogether, I have strong doubt about the reality of such vesicle formation. Better cryo-TEM images should be provided or other evidences based on scattering (SANS, SLS,...) to fully evidence the formation of vesicles, if any.”

Answer:

In terms of the arguments around the interpretation of the SAXS curves based on slope of the log(intensity) versus log(q) plot these arguments largely apply for classical vesicles formed by the self-assembly of lipid molecules. In that case the length-scales associated with the thickness of the bilayer/vesicle and the overall shape of the vesicle are clearly separated and fixed by the packing of the lipid molecules into a bilayer. These results show that with age the particles grow to exhibit the classical -4 slope of flat surface. As time progresses the deviation from the -2 slope increases. Additional experiments have been implemented for clarification on nanoparticle morphology and added to manuscript.

Figure 1A now replaced, staining with uranyl acetate reveal a hollow cavity, suggesting diffusion of dye through a porous membrane structure. Further description has been added to manuscript following from Fig 4. Again, to gain further insight into the capsule morphology if any, we removed curcumin and observed the final nanoparticles under TEM seen in Fig 4C.

2/ *“The other very strong weakness is about the mechanism of formation and stabilization of the nanoparticles. Previous contributions about the formation of vesicles involving strong H-bonding are referred. However, in these systems (ref. 24, 25), H-bonding was occurring in organic solvent. Here, the formation of H-bonding between curcumin and sugar in water is not obvious. It is well known that water is not a favorable solvent to stabilize such interactions (even worse if polar solvents such as DMSO are also present). Temperature or pH variations are not sufficient to prove the presence of H-bonding. Why not adding urea for example?”*

Answer:

Please see following changes to manuscript, “It should be noted that in nature that a variety of highly smart nanomaterial systems are constructed by the association of various kinds of biomacromolecules, including enzymes, proteins and DNA via well-controlled intermolecular and/or intramolecular interactions in water. For this reason, it seems that the weaker hydrogen bonding between the fructose and curcumin hydroxyl groups are sufficient to drive the self-assembly in water.” We have also found higher amounts of DMSO ($> 90 \mu\text{L}$, 4 w/w%) led to polydisperse nanoparticles because of increasing polarity that may not favour hydrogen bonds. This data has been added in **ESI, Table S6**.

The fructose molecules can stabilise and shield the strongly hydrophobic and non-polar segments of curcumin in water into a soluble vesicular aggregate. In addition to hydrogen bonding, hydrophobic interactions between curcumin in water also play a role in this self-assembly reported by Bagchi and coworkers, J. Chem. Phys. (2014) 18:141.

We appreciate this experimental idea to provide evidence of hydrogen bonding. Urea was titrated against nanoparticle solution with interval DLS measurements taken over 30 minutes (see figure below). From our results, the auto-correlation function shows particle instability and gradual precipitation with increasing cumulative additions of urea. Lower intensity % of nanoparticles is influenced by urea concentration and eventual precipitation out of the

nanometer range further supports the loss of nanoparticles through perturbation of the delicate intramolecular and intermolecular hydrogen bonding. This indicates to us of possible hydrogen bonding between fructose and curcumin molecules that can be displaced by urea, although intrinsic hydrophobic driving forces are present to enable tolerance to chaotropic agents alike.

Figure. DLS measurements of nanoparticles prepared by titrating urea solution against [Fru] = 31.25 mg/mL, [CCM] = 60 µg/mL nanoparticles (A) Size-intensity graph (B)Auto-correlation function graph

3/ In addition, sugars are also known to present a hydrophobic character (see for instance: *Hydrophobic nature of sugars as evidenced by their differential affinity for polystyrene gel in aqueous media. Janado, M. & Yano, Y. J Solution Chem (1985) 14: 891*). The sugars could also stabilize the interface of nano-clusters of curcumin by absorption at the interface, due to their hydrophobic character.”

Answer:

Yes, this point follows and links up to #2 from reviewer 2. In Figure S16, S17 and S18, only certain carbohydrates, namely glucose and fructose can form hollow particles instead of nanoclusters. Our observations indicate the importance of the hydroxyl groups positions around the ring, where equatorial positions have higher reactivity than their axial counterparts available for hydrogen bonding to favour the formation of these hollow particles as described in the manuscript. However, hydrophobic driven interactions cannot be ignored, as the tendency for curcumin to agglomerate is strong and accompanied with the non-polar regions of sugars to support that: “sugars can stabilise the interface of the nano-clusters of curcumin by absorption at the interface, due to their hydrophobic character”

4/ “I have a major concern about the DLS measurements. It is not described in the experimental part and ESI how the solvent viscosity is corrected to determine accurately the Rh. Indeed, when sucrose and curcumin (including organic solvent such as DMSO) concentrations are varied, how the changes in dn/dc and viscosity are considered? The size of the nanoparticles is presented to be constant, but is it really the case?.”

Answer:

We understand this concern, minute amounts of DMSO were used, please see ESI, Table S5. Changing the viscosity respective to the DMSO/ water mixture (*J. Chem. Eng. Data* 7, 1, 100-101) did not lead to significant changes to size reported in manuscript. To

confirm the accuracy of the size determination, our SAX data reports a relatively consistent nanoparticle size independent of the fructose concentration. Please check the SAXS data fitting (Table 1), it shows the size of diameter is around 100 nm. However, the samples were placed for more than 24 hours, the size of diameter increase dramatically, then precipitate.

Reviewer 3:

The recommendation of Reviewer 2 is “*This work by Zhao and coworkers characterizes small hollow spheres (vesicles) formed by the relatively non-polar curcumin molecule in fructose solution. The proposed importance of the observation is that it indicates a new type of self-assembled structure — formed of two components that are not covalently linked. The vesicles have diameters of approximately 100 nm and surface thickness of 16 nm. This is characterized by electron microscopy (Figure 1) and dynamic light scattering (Figure 2). Spectroscopy indicates that the electronic environment around the curcumin and fructose changes substantially as part of the vesicle structure (Figure 1) implying they are interacting.*”

Response and revisions:

1/ “*I find the observation of these structures to be interesting and certainly to defy my expectations. My expectation is that curcumin would always form solid particulate assemblies to minimize its apolar surface exposure. In lipid bilayers vesicles this route is not possible because the polar piece is covalently bonded to lipid. Fructose is rather soluble in water. So is something strong happening at the fructose-curcumin interface (analogous to the strong bonds in amphiphiles)? Or, as I prefer below, are these vesicles actually very small bubbles that are perhaps kinetically stabilized by fructose?*”

Answer:

This hypothesis is certainly interesting, we substituted DMSO (soluble in water and dissolves curcumin) with other organic solvents including ethanol, acetone and THF have all found to generate similar outcomes under the same conditions, please refer to **ESI, Table S2**. It is unlikely to form relatively similar nanoparticles with consistent hydrodynamic diameter if they are bubbles. This points responsibility to fructose-curcumin interactions, we herein give evidence with a control experiment by eliminating fructose in the system “**The ability to form vesicles in aqueous solution with and without carbohydrates can highlight the role of strong hydrogen bonding as an integral role in self-assembly. Without sugar, a similar nanoparticle system would not exist merely in water (ESI, Fig S15).**” TEM images in ESI, Fig S15 reveal absence of nanoparticles, instead large precipitates/crystals out of the nanometer range. Insights using SAXS analysis on this matter has been included in manuscript.

2/ “*The paper does not answer this important question. Molecular dynamics simulations were employed to investigate the chemical mechanism of the self-assembly. Potentials of mean force (PMFs) between curcumin and fructose are the key of their simulation “experiment”. These simulations showed that curcumin was more attracted to itself (3 kcal) than to fructose (1 kcal). Kcal is the reasonable energy scale for these interactions. A “control” for fructose-fructose would have completed this picture. So, shouldn’t curcumin simply aggregate according to our expectation? Experiments with other sugars indicate that vesicles aren’t formed. Would PMFs between curcumin and these other sugars indicate a difference?*”

Unfortunately this level of subtlety is likely below the ability of molecular dynamics force fields to help with. In summary, the PMFs, at least superficially, do not indicate that the surface should be stable. Rather, they indicate that it should collapse because the curcumin-curcumin interaction is so strong.”

Answer:..

To better understand the origin of fructose-driven curcumin dissolution, we considered a thermodynamic cycle for the dissolution of curcumin in water (blue) and fructose solution (red):

As shown, differences in the thermodynamic driving force for curcumin dissolution in the two solvent systems could arise as a result of more favourable free energy of solvation of curcumin in fructose solution, $\Delta G_s(\text{ fruc-water})$, and/or a highly exergonic free energy change due to vesicle formation, $\Delta G(\text{ vesicle})$.

The Potential of Mean Force (PMF) simulations of a curcumin molecule immersed in a periodic box of water with and without (200) fructose molecules indicate that fructose enhances the free energy of solvation of curcumin relative to pure water; however, the enhancement is modest, approximately 1 kcal mol^{-1} . Accordingly, this result implies that the thermodynamic driving force for curcumin dissolution in fructose solution must originate from vesicle formation, $\Delta G(\text{ vesicle})$.

DG_{sub} = free energy of sublimation

$DG_s(X)$ = free energy of solvation in solvent X

$DG(\text{ vesicle})$ = free energy change associated with vesicle formation in solution

$$DG_{\text{diss}} = DG_{\text{sub}} + DG_s(\text{ fruc-soln}) + DG(\text{ vesicle})$$

$$DG_{\text{diss}} = DG_{\text{sub}} + DG_s(\text{ water})$$

The time-scale associated with self-assembly is in a matter of seconds, which is beyond the reach of conventional MD simulations. As such, we have constructed a vesicle model which suggests that the vesicle is stable under zero surface tension (see Pt. 7 below).

3/ “Another confusing point was that the thickness of the surface by electron microscopy was 16 nm. This is ca. 10 times thicker than well above the thickness of curcumin. There must be many layers of curcumin in the surface. Is fructose incorporated in the surface center? Why is the surface so thick.”

Answer:

The comment closely relates to reviewer 2's point #1. We would like to refer back to the replaced Figure 1A that shows a TEM micrograph of fructose and curcumin based particles negatively stained with uranyl acetate that reveals a hollow cavity and suggests a porous membrane structure. However, this alone cannot verify whether it is solely curcumin or interplay of curcumin and fructose molecules contributing to the overall membrane thickness. Interestingly, the absence of a clear ring-like outline in cryo-TEM (Figure 2C), indicates a fuzzy interface constructed by small molecules and not a well-defined boundary self-assembled typically seen with lipids and polymers based vesicles. This phenomena was also observed before, where loose membrane structures led to the disappearance of an outside ring (*J. Mater. Chem. A*, 2016, **4**, 12088-12097). Further SAXS studies are given in the paper. The SAXS data(10A) of fresh samples and the fitting parameters show the thickness of the shell is around 30 nm (Table 1). It is difficult to comment with any greater detail as the effects of the diffuseness of the shell interface and dispersity the shells thickness are confounded.

4/ *“To me this points to metastability — that water is trapped inside and that the assembly would prefer to collapse. that is, it is a bubble, not a vesicle. Vesicles do not get popped. If it is a bubble, it is certainly interesting that they are so small and so uniform. It is also appears clear that fructose is acting as a surfactant between curcumin and water, lowering the tension and thus lowering the energy of the bubble.”*

Answer:

Adding more solvent (4 w/w%) leads to disassembly and precipitation, not larger bubbles. We could like to direct the reviewer to ESI, Table S6 on this matter.

5/ *“A key question is whether the surface are stable under zero lateral tension, like that of a lipid bilayer. Here tension may be applied by the Young-Laplace law.”*

Answer: Please see response to point #7

6/ *“Because the assembly is only stable in fructose, the spheres will naturally disintegrate when transferred outside of fructose. This makes there application extremely limited: limited to solutions of fructose. The authors write: “As the self-assembled structure of these small molecules is fragile, the vesicles were captured and preserved by coating with polydopamine”. Experiments that would substantially alter interest in this work would address the stability of the spheres themselves. Under size exclusion chromatography to remove the fructose solution do the particles collapse? Can they withstand any level of sonication? Can they withstand osmotic stress?”*

Answer:

We agree that this self-assembly is limited to certain carbohydrate solutions, but the simple and instantaneous nanoparticle formation can be extended to template polymerisation. In the manuscript, we demonstrate the simplicity of this template for self-polymerisation of dopamine “As anticipated, after coating, the thickness of the bilayer structure increased from ~ 6 nm to 25 nm, suggesting the successful polymerisation of dopamine on the surface of fructose-curcumin nanoparticles bearing a rough topography and reveal a central cavity upon drying (ESI, Fig. S20).”

Advantages of this template in comparison to common templates such as silica particles that often involve complicated purification processes and time-consuming assembly protocol: “In contrast to polymer vesicles that are promising candidates for drug/gene delivery, our study does not require traditional self-assembly of block copolymers that involve tedious synthetic steps. These supramolecular constructs can be adopted in template polymerisation, in which the formation of the template is quick and easily removed after polymerisation while maintaining the structural integrity of polymer formed. This system is biocompatible, easy to prepare, require little organic solvents and present possible routes to be adopted as nanocarriers and/or nanoreactors.” has been added to manuscript to address promising applications of this system other than potential in drug delivery of curcumin.

In terms of stability, these nanoparticles are relatively stable in a range of conditions. Further description in main text as follows “Increasing the temperature resulted in a slight shift in fluorescence maximum toward free curcumin, which suggests reasonably good stability at higher temperatures before signs of disassembly occurs at 45 °C well above room temperature.” and “Osmotic pressure along with gradual removal of fructose cause preassembled nanoparticle aggregation within 3 hours, slight disruption of capsule morphology and steady decline in derived count rate (kcps) can be observed over time (ESI, Fig S11). Thus, stressing the importance of carbohydrate interaction.” In addition, these nanoparticles seem to withstand sonication up to 10 minutes despite signs of less nanoparticles overtime (see figure below).

Figure. DLS measurement of [Fru] = 31.25 mg/mL, [CCM] = 60 µg/mL nanoparticles subjected to sonication over time.

7/ Can you make a Langmuir-Blodgett film of this stuff?.”

Answer:

We thank the reviewer for their concern about applications of these nano-assemblies we are reporting. Given the appropriate substrate and preparative conditions, we believe these particles present potential as Langmuir-Blodgett films with further optimisation studies. A layer of polydopamine coated fructose-curucmin based particles can be observed when a sample of nanoparticle solution is applied briefly to a copper grid with formvar for 5 minutes and excess was blotted with filter paper and let to air dry overnight, please refer to TEM images below.

7/ “The authors cannot simulate the 100 nm diameter particle because the simulation is too large. Can they simulate a planar bilayer of the "material" much as planar lipid bilayers are simulated to represent large vesicles? If the surface is stable at zero tension this would be very interesting. If it collapses (ie, it thickens uncontrollably to reduce its projected area) it may indicate extreme metastability. The tension of the curcumin-water surface with and without fructose could be compared by simulating planar sheets at fixed projected area and reporting the tension as is commonly output in molecular dynamics simulations. Can it be reduced to zero? Current forcefields are not perfectly reliable for the surface tension between polar and apolar compounds but the answers might provide some hints as to what the fructose is doing..”

Answer:

A planar model of the vesicle (composed of 6000 water and 350 fructose molecules on either side of a layer of 200 curcumin molecules) was constructed using PACKMOL. Specifically, the initial configuration of the curcumin layer was constructed such that the curcumin molecules were aligned with the z-axis (surface normal) similar to a lipid bilayer. Upon equilibration, a 200 ns MD trajectory indicates that the vesicle is stable under zero lateral surface tension, i.e. it did not thicken uncontrollably to reduce its projected area. A snapshot of MD trajectory of the vesicle model before (left) and after (right) equilibration is shown below. The simulations also indicate that there is a concentration gradient of fructose molecules from the curcumin layer that is consistent with experimental observations (see below).

To probe the effect of fructose on the stability of the curcumin layer, we have also performed another set of simulations using an identical periodic cell where fructose molecules are replaced by water molecules. Within a 200 ns trajectory, the curcumin layer remains stable under zero lateral tension; however, there appears to be increased disorder in the orientation of curcumin molecules. In particular, we defined an internal vector connecting two carbon atoms along the backbone of curcumin to probe the orientation of these molecules. As shown below, the distance between these two carbon atoms is approximately 2.5 Å, and we monitored the z-component of this vector ('z-value') for all 200 curcumin molecules during the MD trajectory (the z-value is zero when the curcumin molecule is perpendicular to the surface normal). The figure below compares the histogram of z-values for the 200 curcumin

molecules based on the last 50 ns segment of the trajectory. As shown, the fraction of curcumin molecules that have z-values around ± 2.5 Å, i.e. parallel to the surface normal, is significantly lower for the simulation missing the fructose molecules. We have carried out a duplicate (independent) set of simulations and confirmed that this observation is reproducible. Numerical integration of the histograms (averaged over 2 independent trajectories) showed that the fraction of curcumin molecules with $|z| > 2.5$ is approximately 57% and 46% in the presence and absence of fructose molecules respectively.

While the time-scale of these simulations are short compared to experiment, they suggest that fructose molecules may play a stabilising role in the orientation of curcumin molecules that may be important for structural integrity of the vesicle.

8/ “Their presentation of the small angle x-ray scattering (SAXS) data, intended to show a sharp surface of the particles was very difficult to parse. Electron microscopy images of the samples visually indicate that the surfaces are not sharply defined, that in fact the thickness varies fairly dramatically. The SAXS data appear to rebut this. Figure S5 appears to show perfect agreement to Porod’s law. Figure S6, presumably similar data plotted differently, appears to show much noisier data than S5.”

Answer:

TEM shows the surface is not sharp because of the staining.

We just figured out the stability and size of these capsules are dependent on time. The SAXS data in the manuscript were carried after the samples were placed to sit for 12 hours. We also tried to measure the same concentration of fructose with the increasing amount of curcumin, and the SAXS data shows the sample with more curcumin shows worse stability, and the size is much bigger by time. The slope changes from 2.29 to 3.97. Therefore, we think the previous SAXS data in the manuscript showed non-fresh samples (placed more than 12 hours). Please check the SAXS data for the fresh samples and their fitting (Figure 2D), and they confirm the size and the shape of nanoparticles.

We added some sentences in the SAXS section of main text to explain it.

9/ *“For some sentences the english was so poor that the meaning was not clear. “It has been noted that it is the ability of sugars to participate in strong hydrogen bonding, especially when the sugar is in its cyclic form, that is one of the key features in obtaining structures of higher order as this imparts rigidity, such as the fascinating hierarchical structures based on sugar amphiphiles” and “Currently, four different crystalline polymorphs are known, which differ in the planarity of the curcumin molecule and as a consequence, in the position of its most prominent hydrogen bonds, is present in the enol form”.*

Answer: These sentences have been revised and clarified in manuscript as follows:

“It has been noted that it is the ability of sugars to participate in strong hydrogen bonding, especially when the sugar is in its cyclic form. This is among one of the key features in obtaining structures of higher order as this imparts rigidity,⁵ including the fascinating hierarchical structures obtained based on sugar amphiphiles.”

“Currently, four different crystalline polymorphs are known, which differ in the planarity of the curcumin molecule. As a consequence, the position of its most prominent hydrogen bonds is present in the enol form.”

Reviewers' comments:

Reviewer #2 (Remarks to the Author):

The authors have now fully answered the points raised and the manuscript can be accepted as such.

Reviewer #3 (Remarks to the Author):

This revision of the paper by Wong et al is much improved. This work has the potential to influence thinking of how nanometer-scale amphiphiles can be designed to self assemble target structures.

I was able to interpret their analysis of the SAXS data. Their simulation component is much improved, I write about it in detail below. Generally, the new simulations show that a curcumin layer can be stable under zero tension — a required property for it to form a vesicle or capsule. PMFs and histograms suggest a favorable curcumin-fructose interaction that could stabilize the interface further under certain fructose concentrations.

Two pieces are missing that prevent the result from being completely “solid”: First, the authors did not show that curcumin-water alone collapses, or that the properties of divanillin, mannose and galactose is different from curcumin or fructose. Second, their simulated surfaces are much thinner than SAXS indicates. The SAXS surfaces are relatively very thick and may be challenging to simulate. However, thicker surfaces may be necessary to address the first point. A thicker surface may collapse more quickly than the thin surfaces the authors have simulated here.

Regarding the simulations:

PMFs indicating that free curcumin is more stable in fructose versus water provide some nice information pointing away from particle stability. They form a nice companion to the layer simulation shown in Figure 3B. The authors have demonstrated that a curcumin layer is feasible at zero tension. Unfortunately they were not able to show that the layer without fructose “collapsed” to minimize water/curcumin exposure. Their PMF simulations suggest there should be a greater tension between curcumin/water and curcumin/fructose solution.

I would have liked to see the authors report the area per curcumin. Shrinking projected area/curcumin for the water+curcumin layer simulation (relative to w/fructose) would indicate a greater tension between curcumin and water. Simulating the species that did not form stable capsules (divanillin, mannose) would have been particularly illuminating. It makes them appear rather incurious that the authors did not even simulate mannose which is not difficult to parameterize.

The simulated surfaces are much thinner than the observed surfaces. This is a weakness of the modeling. Thicker solutions might have collapsed in the water but not with fructose. The simulated layers can be run through the software “SimToExp” to produce SAXS scattering spectra that will likely show that their simulated bilayer is too thin. The current fit to the SAXS data is of a surface with homogeneous scattering length density. The simulations may indicate a more inhomogeneous SLD (this analysis would be applied at q sufficiently high that the shape of the capsule was not important). Simulating surfaces with the approximately correct thickness (ca. 8 nm) would be very interesting to compare to the experimental SAXS. I believe

From Figure 3D it appears that the curcumin molecules were all started with the same orientation, and that they may very slowly be flipping orientation. To me the figure indicates that the simulation is unequilibrated. Given the molecule itself I don't expect there to be a large bias one

way or the other, up or down. It would have been better if the layer had been formed with the molecules in random orientations.

The interpretation of the enhanced ordering with fructose is definitely suggestive but is difficult to interpret rigorously. Certainly a surface will tend to be ordered, and the constituent molecules of a particle could very well be disordered.

In short, I believe there are a few important unturned stones indicated by these simulations but that this section is much improved.

The simulation writing could be trimmed by focusing on the tension between curcumin and water or water/fructose and downplaying the thermodynamic cycle which I do not think many readers will go to the supplemental to see. I would support the authors highlighting ordering as a liquid-crystalline property near the beginning of the discussion. With regards to the cycle, a major component is left uninvestigated: why the capsule surface is stable. This is due to the benefit of the curcumin-fructose layer which the simulations show nicely in their histogram.

On Figure S12B please indicate whether the water or fructose solution is above or lower than zero.

The authors write "Curcumin, under investigation as a potential drug against a range of diseases..." My non-expert understanding is that studying curcumin as a drug is somewhat controversial. Its use as a drug appears minimally related to its use for forming small capsules. Its properties as a promiscuous drug may even prevent it being the eventual self-assembling species. It is of course the authors' choice but I would restrict commentary to the very interesting physical/chemical aspects of the problem that the paper addresses.

Black = The reviewer comments

Red = Response to reviewers

Responses to Editorial Office and reviewers:

Reviewer #2

The authors have now fully answered the points raised and the manuscript can be accepted as such.

Thank you

Reviewer #3

This revision of the paper by Wong et al is much improved. This work has the potential to influence thinking of how nanometer-scale amphiphiles can be designed to self assemble target structures. I was able to interpret their analysis of the SAXS data. Their simulation component is much improved, I write about it in detail below. Generally, the new simulations show that a curcumin layer can be stable under zero tension — a required property for it to form a vesicle or capsule. PMFs and histograms suggest a favorable curcumin-fructose interaction that could stabilize the interface further under certain fructose concentrations.

Responses and revisions

1/ Two pieces are missing that prevent the result from being completely “solid”: First, the authors did not show that curcumin-water alone collapses, or that the properties of divanillin, mannose and galactose is different from curcumin or fructose.

Answer:

Curcumin-water alone collapses into major crystalline precipitates as shown in the ESI, Fig S15 and referred to in manuscript “The ability to form hollow particles in aqueous solution with and without carbohydrates can highlight the role of strong hydrogen bonding as an integral role in self-assembly. Without sugar, a similar nanoparticle system would not exist merely in water (ESI, Fig S15) and render curcumin insoluble as large precipitates”

In the revised manuscript, the MD simulations (see below response to #4) indicate that the area per curcumin is consistently smaller for the simulations without fructose compared to the simulation with fructose present (see revised Figure 3D). This is consistent with the higher surface tension of the curcumin layer when fructose is absent, suggesting that fructose mediates the interaction between water and curcumin. Also shown in Figure 3D, the curcumin layer is still shrinking even after 100 ns. Presumably, over a longer simulation time, the curcumin-water layer would eventually collapse, as observed experimentally..

We have also performed simulations for curcumin in a mannose solution (which does not result in capsule formation); however, it appears that conventional molecular mechanics force field simulation is unable to resolve the subtle difference between fructose and mannose (see our response to #4).

2/ Second, their simulated surfaces are much thinner than SAXS indicates. The SAXS surfaces are relatively very thick and may be challenging to simulate. However, thicker surfaces may be

necessary to address the first point. A thicker surface may collapse more quickly than the thin surfaces the authors have simulated here.

Answer:

It is unclear at what boundary of the nanoparticle the SAXS detects considering these fructose-curcumin based capsules are ordered by small molecules and do not necessarily have a sharp and well-defined boundary interface as compared to polymersomes, liposomes and surfactant based vesicles. Thereby, it is reasonable to take this point into account that the SAXS surfaces could be thicker and assume it detects highly concentrated areas within the periphery of the aggregates. The simulation by molecular modelling was used in the means of gaining insight into stability and thermodynamics of the system with known information: curcumin (0.42 nm in width and 2.03 nm in length)

Regarding the simulations:

*3/PMFs indicating that free curcumin is more stable in fructose versus water provide some nice information pointing away from particle stability. They form a nice companion to the layer simulation shown in Figure 3B. The authors have demonstrated that a curcumin layer is feasible at zero tension. Unfortunately they were not able to show that the layer without fructose “collapsed” to minimize water/curcumin exposure. Their PMF simulations suggest there should be a greater tension between curcumin/water and curcumin/fructose solution. I would have liked to see the authors report the area per curcumin. Shrinking **projected area/curcumin for the water+curcumin layer simulation** (relative to w/fructose) would indicate a greater tension between curcumin and water.*

Answer: We thank the reviewer for this important suggestion. We have updated Figure 3D to show the area per curcumin for the water+curcumin and fructose-solution+curcumin simulations. As shown, we do observe a smaller area per curcumin for the former where the value is approximately 21.9 Å²/curcumin compared to 24.0 Å²/curcumin after 100 ns. The simulations indicate that the projected area/curcumin is still shrinking after 100 ns. Presumably, over a sufficiently long simulation time, the layer without fructose would eventually collapse, but it is impractical to probe this using conventional MD simulations. Importantly, these results are in accord with the expectation that there is greater tension between curcumin and water.

4/ Simulating the species that did not form stable capsules (divanillin, mannose) would have been particularly illuminating. It makes them appear rather incurious that the authors did not even simulate mannose which is not difficult to parameterize.

Answer: We have also carried out the mannose-solution+curcumin simulation, and after 100 ns, the area per curcumin is comparable to that of the fructose-solution+curcumin simulation (24.1 vs 24.0 Å²). We attribute this to the fact that conventional force fields (CHARMM in this case) is not able to resolve this level of subtlety (as the reviewer alluded to earlier). The area/curcumin data for the mannose-solution+curcumin simulation is also shown in Figure 3D, and we have added this to our discussion on p.10.

5/The simulated surfaces are much thinner than the observed surfaces. This is a weakness of the modeling. Thicker solutions might have collapsed in the water but not with fructose. The simulated layers can be run through the software “SimToExp” to produce SAXS scattering spectra that will likely show that their simulated bilayer is too thin. The current fit to the SAXS data is of a surface with homogeneous scattering length density. The simulations may indicate a more inhomogeneous SLD (this analysis would be applied at q sufficiently high that the shape of the capsule was not

important). Simulating surfaces with the approximately correct thickness (ca. 8 nm) would be very interesting to compare to the experimental SAXS.

Answer:

While this is an excellent idea, and direct comparison between MD simulations and scattering data is becoming a powerful tool, the program SIMPtoExp has been developed with the specific case of the lipid bilayers in mind. By contrast to lipid bilayers, where the thickness of the lamellae is found to have little or no polydispersity both experimentally and in the results of MD simulations, our small angle scattering measurements show a sample with quite a high degree of polydispersity in the thickness of the lamellae. Furthermore we suggest that computationally it would be too costly to simulate large enough a system to capture this polydispersity. The MD simulations were aimed at capturing the short range interactions which responsible for self-assembly.

6/I believe from Figure 3D it appears that the curcumin molecules were all started with the same orientation, and that they may very slowly be flipping orientation. To me the figure indicates that the simulation is unequilibrated. Given the molecule itself I don't expect there to be a large bias one way or the other, up or down. It would have been better if the layer had been formed with the molecules in random orientations.

Answer: In the revised manuscript, all the simulations involving the 'capsule' model were performed for a curcumin layer where the molecules are in random orientations (see updated Figure 3).

7/The simulation writing could be trimmed by focusing on the tension between curcumin and water or water/fructose and downplaying the thermodynamic cycle which I do not think many readers will go to the supplemental to see. I would support the authors highlighting ordering as a liquid-crystalline property near the beginning of the discussion. With regards to the cycle, a major component is left uninvestigated: why the capsule surface is stable. This is due to the benefit of the curcumin-fructose layer which the simulations show nicely in their histogram.

Answer: The section on the thermodynamic cycle has been trimmed. The new simulations indicate that fructose mediates the interactions between curcumin and water (see above).

9/On Figure S12B please indicate whether the water or fructose solution is above or lower than zero.

Answer: Figure S12 has been updated to include the PMF for curcumin dissolution in water, fructose and mannose solutions. It is also explained in the Figure caption that a z-value of 0 of 0 Å refers to the air-solution interface, and a positive z-value indicates that the solute is above the condensed phase.

10/The authors write "Curcumin, under investigation as a potential drug against a range of diseases..." My non-expert understanding is that studying curcumin as a drug is somewhat controversial. Its use as a drug appears minimally related to its use for forming small capsules. Its properties as a promiscuous drug may even prevent it being the eventual self-assembling species. It is of course the authors' choice but I would restrict commentary to the very interesting physical/chemical aspects of the problem that the paper addresses.

Answer:

We would like to include this point to give the audience a brief overview on the significance of curcumin in terms of current uses, advantages and limitations. Although we are aware that it's use as a drug is somewhat controversial, we carefully state "potential drug" gathered from literature. To the

best of our knowledge, curcumin suffers from low water solubility and bioavailability to be approved as a drug by the FDA, yet this compound has not been reported to act as building blocks in self-assembly in literature as of 2018. However, we have added a sentence referring to a critical review that highlights that some of these studies that suggest curcumin's bioactivity have not been carried out with sufficient scientific rigour.

REVIEWERS' COMMENTS:

Reviewer #3 (Remarks to the Author):

They have addressed my points. Thank you for giving the mannose simulation a try. I agree with their explanation of that simulation experiment. The lack of a molecular model for the full thickness curcumin layer is a minor weakness of the paper but not grounds for more study.

Minor: Some writing in the updated text of the paper has some severe grammatical errors.

See lines 161-164 with many problems, the last sentence on line 217 has no verb.

The sentence "We have also performed MD simulations where fructose molecules are replaced with mannose which did not result in capsule formation" could be misinterpreted as indicating the simulations shows that capsules weren't formed.

Black = The reviewer comments

Red = Response to reviewers

Responses to Editorial Office and reviewers:

Reviewer #3

They have addressed my points. Thank you for giving the mannose simulation a try. I agree with their explanation of that simulation experiment. The lack of a molecular model for the full thickness curcumin layer is a minor weakness of the paper but not grounds for more study.

Minor: Some writing in the updated text of the paper has some severe grammatical errors.

See lines 161-164 with many problems, the last sentence on line 217 has no verb.

The sentence “We have also performed MD simulations where fructose molecules are replaced with mannose which did not result in capsule formation” could be misinterpreted as indicating the simulations shows that capsules weren’t formed.

We like to thank the reviewer, we have changed all the minor comments